# The influence of kindness on academics' identity, well-being and stress

K. Kanoho Hosoda[1]*, Mica Estrada[2]

**1** Department of Native Hawaiian Health, John A. Burns School of Medicine, University of Hawai'i, Honolulu, Hawai'i, United States of America, **2** Department of Social & Behavioral Sciences, Institute for Health & Aging, University of California, San Francisco, San Francisco, California, United States of America

* khosoda@hawaii.edu

**Data Availability Statement:** All relevant data are within the manuscript and its Supporting Information files.

**Funding:** KH #K99GM151640 National Institutes of Health https://www.nigms.nih.gov/ ME #3R01GM138700-01S1 National Institutes of

## Abstract

The well-being of people working and studying in higher education, including students, staff, and faculty, is a topic of increasing concern. The lack of well-being may be attributed to the current academic context, which does not consistently provide cues that affirm social inclusion to all members of the academic population. This study examines the role of kindness (defined as actions that affirm dignity and social inclusion) in promoting identification with community and well-being in higher education utilizing a cross-sectional study of 182 diverse members of higher education. To assess the extent that kindness relates to the acquisition of institutional identity, well-being, and stress, we developed and validated two novel psychometric rating scales for kindness: Kindness Received (α = 0.927, ω = .921) and Kindness Given (α = .859, ω = .860). Initial analysis showed that receiving kindness was significantly associated with increased well-being, reduced stress, and improved institutional identity. Giving kindness was significantly associated with decreased stress reduction and decreased institutional identity. Results from structural equation modeling shows that institutional identity mediates the relationship between receiving kindness and well-being. Qualitative analysis of micronarratives regarding kindness showed that feeling safe and being acknowledged are the most commonly described experiences of kindness, both acts that affirm dignity. The findings from this study suggests that kindness contributes towards improving diverse people's well-being and increased identification with institutions of higher education. Measurement of kindness provides methods for assessing institutional changes that foster greater positivity and inclusion in higher education settings.

## Introduction

Well-being is associated with an individual's satisfaction with life as a whole and with improved productivity and worker retention rates [1, 2]. The workplace environment plays a large role in the lives of academics because they spend on average 40 to 55+ hours per week working [3, 4]. In academic settings, the well-being of individuals, including students, staff, and faculty, is a topic of increasing concern. The three pandemics–COVID-19, racism, and climate disasters–have created challenges to well-being and raised awareness that the way we treat each other matters. Members of academia who are historically underrepresented also

Health https://www.nigms.nih.gov/ The funder did not play any role in the study design, data collection and analysis, decision to publish, or preparation of the manuscript.

**Competing interests:** No authors have competing interests.

report experiencing an additional layer of low connection in the form of micro- and macro-aggressions, which are forms of subtle or overt discrimination and prejudice and are shown to hinder social connection and contribute to disparities in retention [5]. While there is a wealth of literature discussing the reasons why people choose to leave the academic community and research describing how global experiences such as bridge programs, research internships, and mentorship programs lead to positive outcomes for students, there is indeed limited research focusing on the specific behaviors that convey cues that affirm inclusion and belonging, particularly for career academics [6–8].

Review of the literature suggests that the current academic context does not consistently provide cues that affirm social inclusion to all members of the academic population equally [9]. The lack of affirmation has consequences and undermines a basic need that people are more likely to survive and experience well-being when feeling socially connected [10–15]. The research suggests that people, including academics, are social beings, and feeling socially connected has a significant impact on their overall happiness and life satisfaction. Further, literature shows that social connection can reduce experiences of stress in pressured environments [16, 17]. Social science research provides evidence that social contextual variables—specifically kindness cues affirming social inclusion—can contribute to increasing retention and persistence of a diverse population in academia [9]. In this study, we define and measure the unique construct of kindness and examine how academics experience kindness in their institutions and to what extent do experiences of kindness relate to academics' experiences of well-being, stress, and self-identification with one's institution.

## Conceptualizing kindness

A variety of definitions of an act of kindness exists in the literature [18, 19]. The emphasis in these definitions is on the nature of the action done. These actions are difficult to differentiate from definitions of helping or altruism. The *Handbook of Social Psychology* [20] does not have a definition of kindness, whereas *Positive Psychology* defines kindness as "doing favors and good deeds for others; helping them, taking care of them" [21]. In this study we use Estrada et al.'s (2018) definition of kindness, as: "an action that results in the affirmation of the dignity of the recipient of the act" [9]. The definition builds on the work of Hicks who eloquently defines dignity as a birthright that relies on "treat[ing] others as if they matter, as if they are worthy of care and attention" [22].

## Kindness cues affirm social inclusion

Kindness cues affirm social inclusion by communicating respect for the dignity of another [23]. Research evidence shows that humans are consistently scanning their environment for social cues to assess danger and safety [24]. The impacts of threatening or aggressive social indicators, including macro- and micro- aggressions, racism and prejudice, are shown to reduce experiences of belonging and inclusion in higher education environments, and can impact health and mental well-being of people from marginalized communities [25]. Research has also shown that violations of one's dignity is a form of rejection, resulting in experiences of social exclusion and at times conflict [22, 26, 27]. In contrast, social affirmations encourage members to be a part of the community, with research focusing on perceptions of smiles [28, 29]. When a person experiences an act that affirms their dignity, they experience social inclusion, respect for one's life and acceptance of their identity [30]. While dignity is a well-used concept, measurement of kindness, as actions that affirm dignity is new.

Hicks' (2011) dignity theory and research indicates that acts of kindness that affirm dignity would results in the receiver of kindness experiencing the following: 1) acceptance of identity,

2) recognition of efforts and talents, 3) acknowledgment, 4) a sense of inclusion, 5) feeling of physical and psychological safety, 6) being treated fairly, 7) autonomy, 8) feeling understood, 9) being given the benefit of the doubt, and 10) being apologized to when one's dignity is violated. In this paper, we build on this approach, and focus this study on kindness cues that affirm the dignity of others in academia.

## Kindness, prosocial behaviors, altruistic actions

Kindness is a unique concept that differs from helping and altruism. Social psychology writing on kindness has focused on helping others through prosocial behaviors and altruistic actions, analyzing kindness from the perspective of actions done by the agent (i.e., the actor) and how the agent feels when a task of kindness is completed [18, 31]. Prosocial behaviors are numerous and described as positive, but do not address how the receiver feels and the impact of the kind action on the receiver's sense of community and desire to remain in academia. Similar to studies on prosocial behaviors, research on altruism has focused on the agent of kind acts. Studies of altruistic acts found agents choose to conduct acts of altruism to promote their personal values and identity [18, 32]. Prosocial behavior and altruism are defined by the agent of an act, whereas kindness is defined by the receiver of the act [23]. The receiver of the act of kindness may interpret the act of kindness differently, or not at all, in comparison to the agent of kindness' intentions [33]. With this approach, acts of helping or altruism are only kind actions when the receiver experiences an affirmation of their dignity. In this study we aim to understand how kindness is perceived from the receiver when addressing the research question: How do academics experience kindness in their institutions?

## Connection to institutional identity, well-being, and stress

Kindness, in the context of work environments and academia, may have impacts beyond being a pleasant experience. Kindness inherently has impacts on the social experience and may impact people's connection to the institutions in which the person works. One strong measurement of social connection is the strength of identification with a group, community or institution, which can satisfy a person's innate need for community and affiliation with others. Identification with one's academic institution, for instance, has been found to contribute to greater integration and promote retention and persistence [12, 13, 15, 34–36]. The human desire for connection is so strong that when a person feels disconnected from a social group it is processed by the brain in a manner similar to physical pain [14]. Research on student integration into scientific communities found that self-identifying as a scientist (i.e. feeling as though they are a part of the community of scientists) is more uniquely predictive of science community integration and persistence than science efficacy (having the confidence to do science) alone, resulting in students engaging in behaviors and expectations consistent with the role of a scientist [26, 37]. We recognize that some literature conflates self-identifying as a member of academia with a sense of belonging to academia. However, self-identity is present when "an individual accepts influence from another person or a group in order to establish or maintain a satisfying self-defining relationship to the other" [27] whereas sense of belonging exists when there is an "experience of personal involvement in a system or environment so that a person feels themselves to be an integral part of that system or environment" [38]. We surmise sense of belonging is an attribute that can contribute to identification, where one may feel as though they belong in academia, but it is possible they do not self-identify as a member of the academic community and regularly challenge the norms of the community. Identification as a member of the academy, meaning one's sense of self is drawn from affiliation with a group, is a strong complex social influence process [37].

In addition to institutional identity, well-being, and stress are attributes that are associated with academic success [39]. When people experience the combination of high well-being and low stress, people are more likely to persist. Life satisfaction is one of three components of subjective well-being, the others being positive affect and negative affect, that are based on personal judgments of life quality compared to individual standards of life [1]. Life choices that promote high well-being are associated with continued success, happiness, and maintenance. Conversely, environments that induce stress, including feeling disconnected to a social group, are associated with the fight or flight response, manifested in the academy as dropping out or leaving an institution [11]. Social experiences, such as kindness, could theoretically impact any or all of these factors. We therefore explore the connections between kindness, institutional identity, stress and well-being in this study to answer our second research question: To what extent do experiences of kindness relate to academics' experiences of well-being, stress, and self-identification with one's institution?

## Current study

This study seeks to advance the science of kindness by examining (a) how academics experience kindness in their institutions and (b) To what extent do experiences of kindness relate to academics' experiences of well-being, stress, and self-identification with one's institution. The present study develops and applies two novel psychometric scales of kindness through operationalizing the ten essential elements of dignity theorized by Hicks [22] and contextualized as a psychometric scale by Estrada et al [9]. We hypothesize academics experience kindness when they receive actions that affirm their dignity. This is tested through confirmatory factor analyses of the two kindness scales.

We incorporate validated Kindness Received and Kindness Given scales into a structural equation model used to test hypotheses on the relationships between receiving kindness, giving kindness, institutional identity, well-being, and stress. We hypothesize receiving kindness and giving kindness are positively associated with reduced stress and increased well-being as mediated by institutional identity (see conceptual model presented in Fig 1). We further explore the relationship between receiving kindness and giving kindness through qualitative analysis of a reflective micronarrative on an experience of kindness. We provide this further analysis to better understand the quantitative findings and describe how experiences of kindness conjure positive mental states of mind.

## Materials and methods

### Procedure

The Social Influence of Kindness Study was conducted through a one-time anonymous 10-minute online survey via Qualtrics. The study was reviewed and approved by the University of California San Francisco Mount Zion Committee Institutional Review Board (#21–35884). Written informed consent was obtained from all study participants. Participants were recruited using the snowball method via email and social media that targeted academics in higher education who had interest in taking a 10-minute survey about kindness and dignity. Recruitment and surveys for the study occurred between February 22nd through March 31st, 2022. Following an online protocol, participants first engaged in writing a micronarrative following the instructions:

> We are collecting stories. Think about a time when someone was kind to you in your academic and/or professional life. Take 2 or more minutes to describe this experience. Details

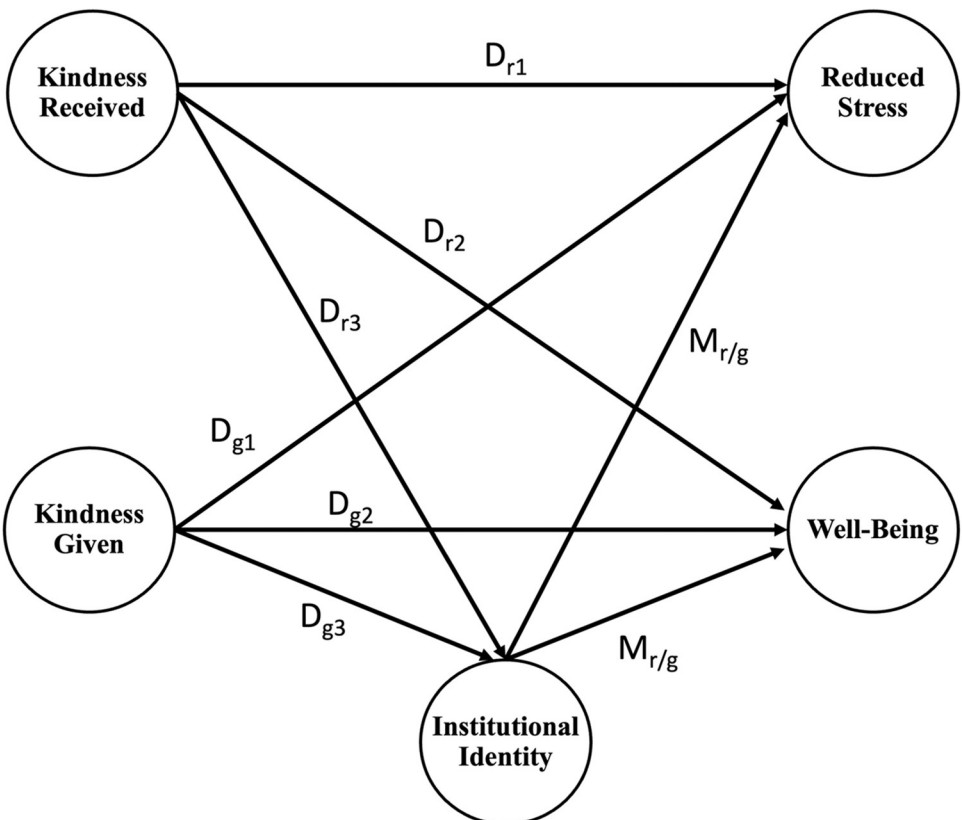

**Fig 1. Conceptual model being tested in the current study.** The paths labeled Mr/g test the mediation effects of institutional identity between kindness received/given on reduced stress and well-being. Paths labeled Dgx test the direct effect of giving kindness on the designated output concept. Paths labeled Drx test the direct effect of giving kindness on the designated output concept. All tested paths are hypothesized to have a positive association with outcome concepts.

are welcome regarding what happened, how it felt, and why this moment is memorable to you.

After writing the micronarrative they then were asked to list 3 words that described how they felt after recalling the experience of kindness. Finally, they completed a series of measures described below.

## Participants

For this study, a subset (N = 182 of 215) of the larger Social Influence of Kindness Study were selected for analysis based on their occupation designation of studying or working within an academic institution. The sample of 182 included 77.47% working in academia and 22.53% studying in academia; 21.43% in fields of STEM as defined by the National Science Foundation's categories and 78.57% working in non-STEM fields in academia; 69.23% female, 17.58% male, 3.85% non-binary, and 9.34% not reporting gender, see Table 1. The ethnic demographic distribution included 53.85% White, 14.29% Hispanic, 13.19% Native American, Alaska Native, Native Hawaiian, or Pacific islander, 8.24% Asian, 4.40% African American, and 6.04% not reporting ethnic background. In line with the National Institutes of Health's (NIH) categorizations in 2020, underrepresented racial minorities (URM) were defined as those who were

**Table 1. Summary of sample descriptive statistics.**

| Variables | N | M | SD | Skew | Kurtosis |
|---|---|---|---|---|---|
| Gender (1 = Male, 2 = Female, 3 = Non-Binary, 4 = Prefer not to answer) | 174 | 1.97 | 0.68 | 1.39 | 4.20 |
| Occupation (1 = Work in Academia, 2 = Student in Academia) | 182 | 1.23 | 0.42 | 1.33 | -0.24 |
| Underrepresented Minority Status (1 = No, 2 = Yes) | 182 | 1.32 | 0.47 | 0.76 | -1.44 |
| Institutional Identity | 181 | 3.75 | 1.06 | -0.72 | -0.28 |
| Well-Being | 179 | 3.68 | 0.77 | -0.46 | 0.02 |
| Stress Level | 181 | 3.28 | 0.38 | 0.14 | 0.49 |
| Kindness Received | 181 | 3.56 | 0.62 | 0.08 | -0.01 |
| Kindness Given | 181 | 4.16 | 0.46 | -0.32 | -0.34 |

M = mean. N = sample size (cases with complete data for a given variable). SD = standard deviation. Institutional Identity, Well-Being, Stress Level, Kindness Received, and Kindness Given variables data are for the intercorrelations.

African American, Hispanic, Native American, Alaska Native, Native Hawaiian, or Pacific islander, for a combined URM sample of 32.42%.

## Measures

All scales were administered via anonymous self-report online surveys. Questions asked participants to reflect on experiences, feelings, and thoughts in the past month, unless otherwise stated.

**Kindness given and received.** The Kindness Given scale measured how often in the past month the survey participant engaged in actions of kindness through affirming another person's dignity (e.g., "treated others fairly"). The Kindness Received scale measured how often the survey participant experienced kindness from another person through actions that affirmed their own dignity (e.g., "Your efforts, thoughtfulness and/or talents were positively recognized"). Both the Kindness Given and Kindness Received scales comprised 10 items developed from Hicks' essential elements of dignity items shown in Table 2 [22]. Each item on the Kindness Given and Kindness Received scales had five response options ranging from 1 (never) to 5 (every time). Scale scores were derived as an average of the items, with higher scores indicating more frequent experiences of kindness. Since these are novel measures, the 10 items from each of the Kindness Given scale and the Kindness Received scale were predicted to load on to one factor based on the theory that affirmations of dignity promote kindness [9].

**Institutional identity.** This is a three-item scale modeled after Estrada et al.'s [26] *Science Identity Scale* that was used to define the extent to which participants perceived themselves as members of their institution that they work for or attend school at (e.g., "I have come to think of myself as a member of the institution in which I study or work"). Participants rated their agreement from 1 (strongly disagree) to 5 (strongly agree). Institutional identity scale scores were derived as the average of three items, with higher scores indicating a stronger institutional identity. Prior evidence indicates that measures of identity are related to persistence within an academic community [37, 40]. The measure is internally consistent in this study (α = 0.90).

**Stress.** This four-item scale is a reduced version of Cohen, Kamarak, and Mermelstein's [41] 14-item perceived stress scale. This is a widely used measure to evaluate the degree to which life events are perceived as stressful. Participants rated their perceived stress in the past month (e.g. "Felt that things were going your way"). Each item had five response options ranging from 1 (never) to 5 (often). Stress scale scores were derived as an average of the four items, with higher scores indicating more reduced stress levels. The current study found evidence of

**Table 2. Ten essential elements of dignity with the associated items used in the kindness given and kindness received scales.**

| Essential Element of Dignity | Kindness Given Measure Reliability: α = 0.859 ẇ = 0.860 | Factor Loading | Kindness Received Measure Reliability: α = 0.927 ẇ = 0.921 | Factor Loading |
|---|---|---|---|---|
| Benefit of Doubt | Gave others the benefit of the doubt | 0.50 | You were given the benefit of the doubt | 0.71 |
| Autonomous | Respected others' freedom of choice | 0.51 | Your choices were respected | 0.78 |
| Apologized to | Apologized when you violated others' dignity in some way | 0.56 | You received an apology when your dignity felt violated | 0.66 |
| Understood | Made an effort to understand others' point of view | 0.57 | Others made an effort to understand you | 0.77 |
| Recognized | Positively recognized others' efforts, thoughtfulness and/or talents | 0.59 | Your efforts, thoughtfulness and/or talents were positively recognized | 0.76 |
| Included | Conveyed inclusion (e.g., in your family, team, club, community, profession, etc.)? | 0.60 | Others conveyed you were included (e.g., in family, team, club, community, profession, etc.) | 0.74 |
| Treat Fairly | Treated others fairly | 0.64 | You were treated fairly | 0.75 |
| Acknowledged | Acknowledged the validity of others' feelings, concerns and experiences | 0.66 | Your feelings, concerns and experiences were acknowledged as valid | 0.80 |
| Safe | Acted in ways that made others feel safe around you (as opposed to being threatening) | 0.73 | Others actions made you feel safe with them | 0.77 |
| Accept Identity | Gave others the freedom to express their authentic selves without fear of you negatively judging them | 0.78 | Felt free to express your authentic self without being negatively judged | 0.66 |

Factor loadings for each scale item are included and the overall scale reliability.

acceptable psychometric properties for the four-item reduced version of the 14-item perceived stress scale (α = 0.71).

**Well-being.** A modified four-item Satisfaction with Life Scale [1] was used to measure perceived well-being. Participants rated their agreement from 1 (strongly disagree) to 5 (strongly agree) with statements regarding their overall well-being (e.g. "The conditions of my life are excellent"). Well-being scale scores are an average of the four-items, higher scores indicate greater well-being. The measure was internally consistent in this study (α = 0.81).

## Data analytic plan

**Scale reliability and validity.** Using the statistical software package Mplus v8.5 [42], we examined the dimensionality of the Kindness Action scale and the Kindness Received scale for those working or studying in academia. In Mplus a confirmatory factor analysis was used to evaluate scale quality. Factor loadings above 0.50 were considered to load on a single factor [43]. The following fit indices were used to evaluate model fit: χ2 statistic, Standardized Root Mean Square Residual (SRMR), Root Mean Square Error of Approximation (RMSEA) and, Comparative Fit Index (CFI). Values representing good model fit are CFI ≥ 0.95, RMSEA ≤ 0.06, SRMR ≤ 0.08 [48]. A Cronbach's α value and a McDonald's ẇ value of 0.80 or greater were used to define the good internal reliability of the scales [46, 47].

**Assessment of model fit in SEM.** Inter-variable relationship analysis was conducted in a structural equation modeling (SEM) framework using *Mplus* v8.5 using the fit indices designated above [42].

**Open ended question content analysis.** Content analysis was conducted to analyze how the academics describe experiences of kindness in their institutions in the micronarratives. The open-ended responses were coded using the ten dignity affirming acts, adapted from Hick's writing on Dignity: 1) acceptance of identity, 2) recognition of efforts and talents, 3) acknowledgment, 4) a sense of inclusion, 5) feeling of physical and psychological safety, 6) being treated fairly, 7) autonomy, 8) feeling understood, 9) being given the benefit of the doubt, and 10) being apologized to when one's dignity is violated. Three coders individually

coded each of the 182 participants' responses, for each of the 10 dignity affirming acts as present or not. Reconciliation occurred between the three coders if there was a discrepancy in the coding. After reconciliation, a response was considered to have attributes of the experience of kindness if at least two of the three coders marked its presence. We recognize the centrality of a researcher in qualitative coding, therefore provide a brief description of the positionality of the three coders. All three coders are female, two of the coders identify as underrepresented minorities, one coder is an undergraduate student, one is staff, and the last is a researcher.

**Immediate recall word frequency.**   To distill how the survey participant felt after reflecting upon the kind action, the micronarrative writing task was immediately followed by a prompt to list 3 words describing how they felt after recalling the experience of kindness. The three words that first came to mind, after they completed writing a micronarrative on an experience of kindness, was analyzed to understand how reflecting on an experience of receiving kindness influences an individual's affect. All words were categorized into positive or negative affective tone using the *bing* and *nrc* sentiment lexicon datasets in the R tidytext package [44, 45]. A word was considered positive or negative if the word was found in either of the sentiment lexicons. If a word was not found in the lexicon, it was not considered for affective tone.

**Gender differences in conducting acts of kindnes.**   The gender of the agent of kindness, described in the micronarrative, was analyzed to understand the differences in which gender is more likely to be recalled as conducting acts of kindness. Upon reading the written reflective micronarratives of kindness, it was noticed that the survey participants often included gender pronouns in their written reflections. We acknowledge that pronouns do not directly correlate with gender, but for this study we chose to use pronouns to categorize the receiver of the kind act's perception of the gender of the agent of kindness. Gender was categorized as female (she/her/hers), male (he/him/his), or nonbinary mention (plural pronouns/proper nouns/no pronouns).

## Results

### Novel kindness scales

Prior to testing mediation models and intervariable relationships, we tested the factor structure and reliability of the Kindness Action and Kindness Received scales.

**Kindness given scale.**   A confirmatory factor analysis was used to analyze the scale fit in a one-factor model. The one-factor model exhibited adequate fit CFI = 0.94 and SRMR = 0.05, although the RMSEA = 0.07 was slightly higher than the cut off of 0.06 we proceeded with the model. The factor loadings were acceptable indicators of the Kindness Given scale (loadings 0.50—to—0.78), as shown in Table 2, thus we concluded the one-factor model provides adequate evidence of measurement validity for use of these items as a scale in this sample of academics. The reliability of the Kindness Given scale was found to be $\alpha$ = 0.86, $\acute{\omega}$ = 0.86 95% CI = (0.83, 0.89) showing this is scale is reliable for the students and workers in academia [46, 47].

**Kindness received scale.**   A one-factor model was a good fit for the Kindness Received scale CFI = 0.979, RMSEA = 0.057, SRMR = 0.031, all of which were within the cutoff scores indicated by [48]. The loadings (0.66-to-0.80), shown in Table 2, were all acceptable indicators for the one-factor Kindness Received Scale. These results indicate a reliable 10-item scale used to measure Kindness Received. The reliability of the Kindness Received scale within this sample population of people who study or work in academia was found to be reliable $\alpha$ = 0.93, $\acute{\omega}$ = 0.92 95% CI = (0.90, 0.94).

### Intercorrelations

Based on a review of literature on kindness, identity, stress, and well-being, we sought to understand how receiving kindness, conducting kind acts, institutional identity, stress and

**Table 3. Intercorrelation matrix of kindness, institutional identity, stress, and well-being.**

|  | Institutional Identity | Well-Being | Reduced Stress Level | Kindness Received | Kindness Action |
|---|---|---|---|---|---|
| Institutional Identity | - |  |  |  |  |
| Well-Being | 0.28** | - |  |  |  |
| Reduced Stress Level | 0.22** | 0.49** | - |  |  |
|  | N = 181 | N = 179 |  |  |  |
| Kindness Received | 0.45** | 0.35** | 0.36** | - |  |
|  | N = 180 | N = 178 | N = 180 |  |  |
| Kindness Action | 0.30 | -0.02 | 0.06 | 0.28* | - |
|  | N = 180 | N = 178 | N = 180 | N = 181 |  |

*$p < .05$.

** $p < .01$.

well-being were related. We conducted a preliminary analysis to understand the general relationships between experiences of kindness, institutional identity, stress, and well-being by examining their intercorrelations. We found receiving kindness is significantly correlated ($p < 0.01$) with institutional identity, reduced stress, well-being, and conducting acts of kindness. In addition, we found that conducting acts of kindness is significantly correlated with receiving acts of kindness, as seen in the correlation matrix in Table 3. These results suggest that experiencing acts of kindness, specifically receiving acts of kindness relates to institutional identity, reduced stress, and well-being.

## Structural equation models

Having found correlations that were consistent with our expectations, we conducted SEMs to understand how receiving kindness and conducting kind acts relates to stress and well-being as mediated by institutional identity, Fig 1 illustrates our conceptual model used to test the relationships between items. The hypothesized model showed adequate fit ($\chi^2$ (435) = 2722.79, CFI = 0.97, RMSEA = 0.03 with 90% C.I [0.02,0.04], SRMR = 0.05). Results from the structural equation model are presented in Fig 2. Consistent with the correlation patterns described, only receiving kindness had significant and positive correlations with institutional identity, reduced stress, and well-being. Interestingly, conducting acts of kindness had a significant negative relationship with institutional identity and reduced stress. Furthermore in this SEM, institutional identity was not a significant mediator between kindness and reduced stress or well-being.

We further examined the effects of giving kindness and receiving kindness as individual constructs in relationship to reducing stress and well-being mediated by institutional identity. We found that giving kindness did not have significant relationships with institutional identity, reduced stress, nor well-being, but interestingly the relationships were all negative, as seen in Fig 3. We found strong positive relationships between receiving kindness with reduced stress and well-being, shown in Fig 4. Additionally, we found that institutional identity mediates receiving kindness and well-being.

## Micro-narrative qualitative analysis

Through the SEM and intercorrelations we found receiving kindness is associated with improved well-being and reduced stress, we further sought to qualitatively understand how receiving kindness is most often experienced in academia at institutions. Coding of micronarratives on kindness (which participants completed prior to answering any quantitative question regarding kindness, institutional identity, well-being, or stress) found that the most

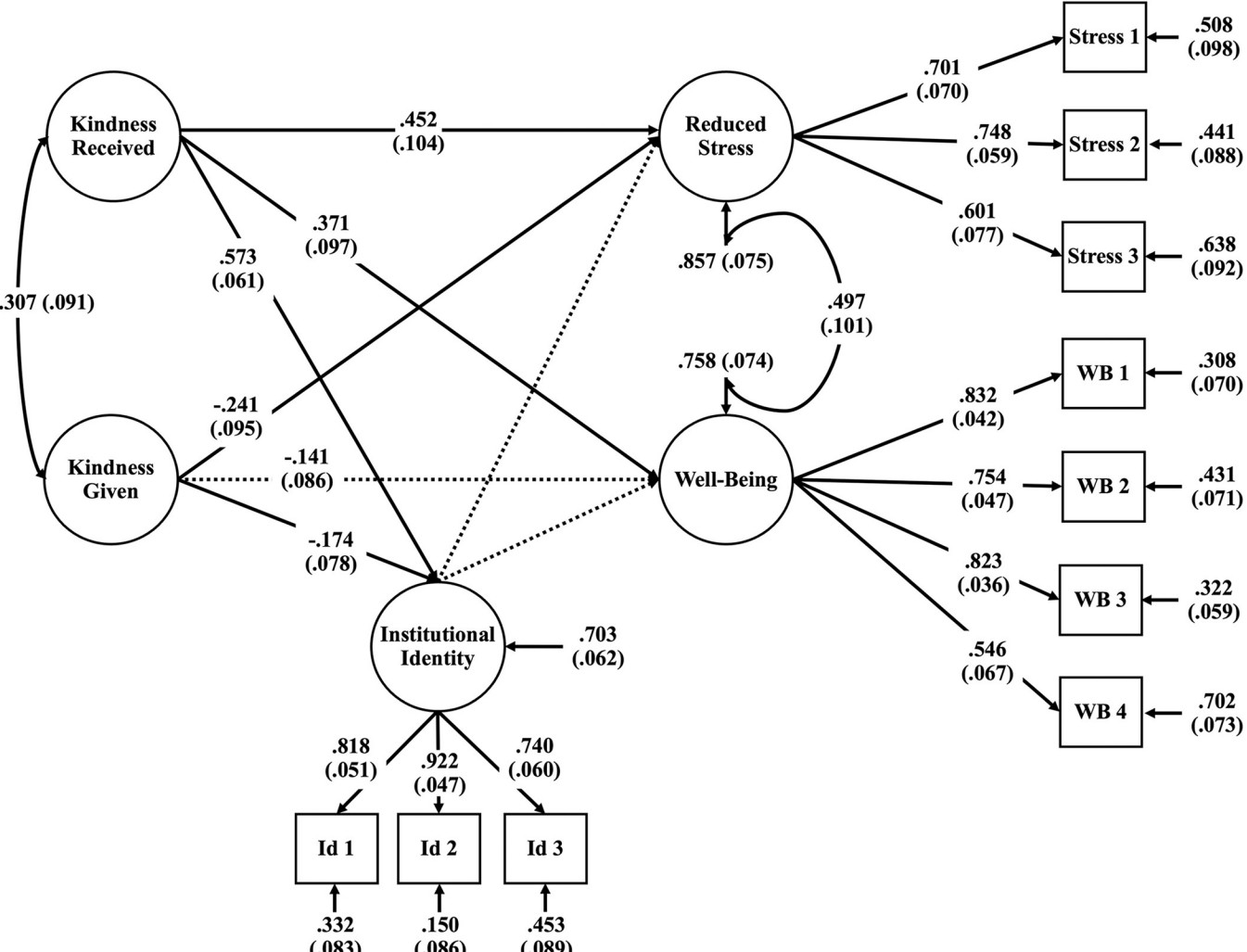

**Fig 2. Structural equation model: Receiving kindness, giving kindness, reduced stress and well-being mediated by institutional identity.** Parameter estimates with standard error designated in parentheses. The solid lines indicate significant paths (p <0 .01). The dotted lines indicate non-significant paths. All paths are standardized.

common way (54.40%) participants received kindness was through *feeling safe*. Participants provided situations where the agent of kindness made them feel safe physically or emotionally.

> I am on a time-intensive committee with two colleagues who are intentional about being supportive in the work each of us does for the committee. . .I feel safe and supported in our group. (Anonymous participant)

*Acknowledging the validity* of the survey participant's feelings, concerns, and/or experiences was described in 53.85% of the micronarratives, describing how kindness was experienced. These responses often included narratives of ensuring that a participant's feelings of self-doubt, or concerns about a situation were addressed.

> As someone with disabilities, I had a very validating moment with one of my professors who shared a personal anecdote saying that she really hopes I do not feel incompetent for being a minority by not only ethnicity, but also disability. Given the lack of representation

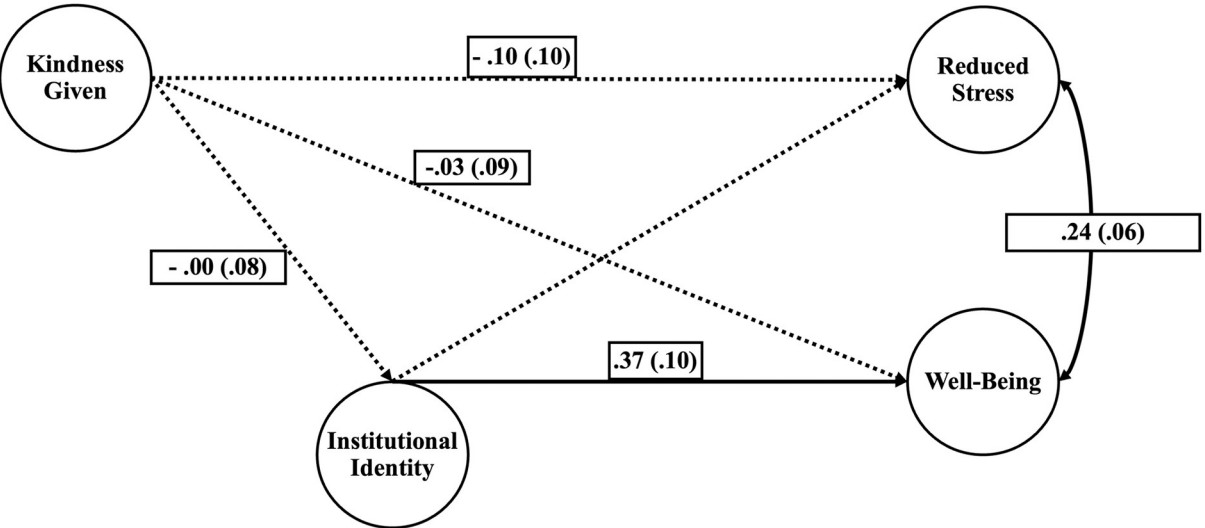

**Fig 3. Model effects of giving kindness on reduced stress and well-being mediated by institutional identity.** Parameter estimates are included with standard error designated in parentheses. The solid lines indicate significant paths (p < 0.01). The dotted lines indicate non-significant paths. All paths are standardized.

of STEM students with disabilities, this felt especially meaningful because it was during my first semester at a new university. (Anonymous participant)

The third most common way kindness was experienced, included in 50.55% of the written responses, is through *positive recognition for efforts*, *thoughtfulness*, *and/or talents*

A professor that I really looked up to talked really highly about me to a conservation group and got me a meeting with them and the leaders of that agency to discuss about my ideas and future collaborations. It felt really validating and decreased my imposter syndrome to

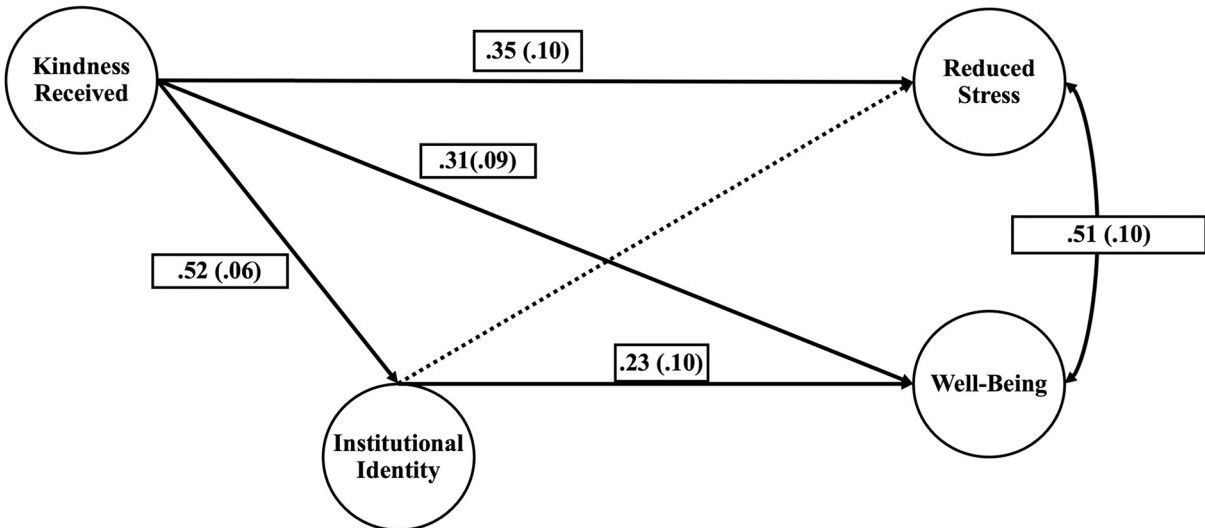

**Fig 4. Model effects of receiving kindness on reduced stress and well-being mediated by institutional identity.** Parameter estimates are included with standard error designated in parentheses. The solid lines indicate significant paths (p < 0.01). The dotted lines indicate non-significant paths. All paths are standardized.

feel that someone as intelligent and well respected as this professor would think so highly of my ideas. (Anonymous participant)

The micronarratives on kindness clearly describe acts in which their dignity was affirmed. Interestingly, from the perspective of the receiver, no responses were coded for experiencing kindness through *apologies*. Specifically, there were no cases in which the agent of kindness apologized to the survey participant when the survey participant's dignity was violated. We surmise this could be due to the survey conditions, where the participants recalled their most memorable experience of kindness. This suggests that apologies after dignity violations are not the most readily recalled kindness experiences. S1 Table shows examples of affirmations of dignity for all 10 categories that were coded for as portrayed through the written responses. The results of the open-ended reflective narrative prompt show that feeling safe, acknowledging the validity of one's feelings, concerns, and experiences, and positively recognizing one's efforts are the three most common ways kindness is experienced and recalled by members of academic institutions.

### Receiving kindness and positivity

We examined the three words that first came to mind after recalling an experience of kindness to better understand how reflecting on an experience of kindness relates to a positive affect. Notably, 89.01% of the participants (N = 162 of the 182 study participants) listed words with positive affective connotations as they reflected how they felt after recalling their experience of kindness as determined by sentiment analysis lexicons [44, 45]. The most frequent words that came to mind were grateful, happy, and warm, all of which have a positive connotation. The second and third words that came to mind for survey participants were also positive, including supported and thankful. Overall, the most frequent words mentioned were grateful (41.98%), happy (22.22%), supported (12.96%), thankful (11.73%), and warm (11.11%). There were words listed with negative connotations like sad, tired, and disappointed but the frequencies were all below 2.00% of all words mentioned. One participant explained why they chose sad, "sad because I'd love to still be working with them" which shows that words with negative connotations may be positive within context. Overall, these results suggest that people have a positive affective experience post-reflection of an act of kindness.

### Gender differences in conducting acts of kindness

Upon reading the reflections of kindness, we noticed that the genders' pronouns were written into the reflections. We hypothesized females would be more frequently recalled as the agent, giver, of kindness, based on research showing female mentors provide more psychosocial support than male mentors [49]. When normalized for the survey participant gender, since 69.23% of the sample population was female, there was no difference in perceived gender of the agent of kindness with 25.27% recalling male kindness agents, 26.92% recalling female, and 47.80% not identifying the agent's gender. We also examined the gender relationship between the agent of kindness and the receiver of kindness. We found females recall acts of kindness by both genders equally, females 29.37%, males 25.40%, no gender identified 45.24%. Males recalling the gender of agents of kindness were similar to the females in that both genders were equally identified, females 21.88%, males 25.00%, no gender identified 53.13%. This finding suggests no perceived gender differences when recalling agents of kindness.

### Discussion

Prior to this study, the concept of kindness was loosely defined in the social psychology literature [21]. Previous studies on kindness conflated the concept with prosocial benefits such as

helping and altruism [18, 31, 50, 51]. This study defines kindness through the development and application of two novel psychometric measures of kindness, the Kindness Received scale and the Kindness Given scale. The kindness scales advance previous work on theoretical approaches that defined kindness as acts that affirm dignity operationalizing, with some modification, Hicks' [22] 10 essential elements of dignity. Our study provides evidence of the positive social influence of kindness in academia through establishing the connection between kindness, institutional identity, well-being, and stress. Furthermore, we describe how academics experience kindness through the lens of dignity affirming actions and extent to which experiencing kindness relates to engaging in acts of kindness. Taken together, this study provides evidence of the importance of kindness in promoting social connection and well-being, an environment in which academics would choose to stay.

## Advancing the science of kindness

Our study defines kindness as an act that affirms dignity; allowing us to directly measure ways kindness is experienced. Our confirmatory factor analyses of the Kindness Received and Kindness Given scales validate the use of dignity affirming acts as constructs of the latent variable kindness. The 10 essential elements of dignity as constructs of kindness provide clearly delineated vocabulary to articulate how kindness is experienced and potentially provide a list of trainable actions for how to increase kindness in academic institutional environments. Delving further into how dignity affirming acts are enacted, we found kindness received amongst academics is most commonly described as occurring through actions that make others' feel safe and acknowledged. This finding suggests that kindness responds to a primal instinct for social connection and the results indicate kindness related to well-being and less stress is consistent with previous research showing people are more likely to survive and prosper when feeling socially connected [10, 11].

## The importance of kindness for social inclusion in academia

Our study establishes the importance of people receiving kindness cues that affirm social inclusion for persistence in academic professions. We provide evidence that receiving kindness is correlated with higher institutional identity, well-being, reduced stress, and giving kindness. These findings are consistent with the work of Chemers [37] and Antaramian [39] highlighting the importance of institutional identity for well-being and its role in academic persistence. This work contributes to the less explored area of research focusing on why academics choose to stay in higher education, which is converse to the majority of previous work on persistence in academia that focused on why academics choose to leave higher education [7, 52]. Academia can benefit from people conducting dignity affirming acts providing more opportunities for academics to receive kindness which promotes social inclusion, through the development of institutional identity, and can improve academic persistence and reduced turnover.

## Receiving kindness, positive self-concept, & further giving kindness

Experiencing kindness promotes inclusion for a scholar to survive and prosper within the academic community. The continuity of kindness within the academic community as a communal value relies on the internalization of kindness as a part of an individual's personal and social identity [53]. We sought to understand the extent to which experiencing kindness relates to self-concept and further conducting acts of kindness. We found that reflecting on acts of kindness conjures positivity. Promoting a positive self-view re-affirms a sense of self which studies have shown is associated with reduction of stresses of academics [54]. Our findings further self-affirmation theory that asserts stresses to one portion of self-concept can be

counterbalanced by affirmations, experiences of kindness, to other portions of sense of self [55]. We found the simple act of reflecting on an experience of kindness promoted positivity through affirming one's self-concept. Harris and Orth's work showing positive self-concept influences positive interactions and relationships with others is quantitatively evidenced in our work as receiving kindness correlates with conducting acts of kindness [56]. Kindness promotes both a positive self-concept and conducting acts of kindness. Interestingly, we found giving kindness was negatively correlated with institutional identity and reduced stress. This may seem counterintuitive as being kind would suggest that one is promoting social connection. However, people who are "too kind" or constantly being the "kind one" become fatigued as the relationship is not reciprocal; they do not receive kindness as much as they give kindness. Balance theory would suggest that when people give more than they feel they receive, social connection can falter [57]. These findings do support the theoretical approach that perceptions of received kindness are critical to kindness occurring as opposed to how kind a person thinks they may or may not be. This finding and the phenomena of the burden of kindness warrants future research.

## Caveats

The present study has several caveats. First the study design was a cross-sectional study that used the snowball method of sampling to collect data. This approach to data collection is ideal for this initial study to advance the science of kindness but should be improved upon in future studies using longitudinal and experimental design methods. We acknowledge that increased variability may exist in this sample of academics, which includes both higher education students and people who work in higher education from various fields of study. We recognize there is a power hierarchy between students, staff, and faculty members in higher education that may influence outcomes. While this study did not directly address the power hierarchy within higher education, future studies would benefit from comparing these different communities within higher education environments. Undoubtedly there are many differences that were not accounted for, such as disciplines, roles, years in academia that may be associated with different experiences in the academy and these differences may be worthy of examination in future studies. An additional caveat is that we worked solely with self-report data. While we used many validated measures that have been shown to relate to real behavior, this study did not measure meaningful outcomes such as graduation or work retention. To make a more direct association between kindness and willingness to stay in academia follow up studies that include retention data and career progression should be included in the analysis. Additionally, findings regarding kindness and its relationship to institutional identity, well-being, and stress would benefit from data collected in non-academic environments to assess the robustness of this initial study. While this study provides a needed first step in measuring Kindness and starting to understand how it impacts people in academia, additional experimental longitudinal studies would be ideal for future study on the influence of kindness to promote connectivity, inclusion, and willingness to stay in academic institutions.

## Conclusion

This study extends the current science of kindness by providing a reliable measure showing that academics who receive kindness are more likely to have strong institutional identity, well-being, reduced stress, and conduct acts of kindness towards others. Further, the results showed that receiving kindness and giving kindness can be reliably measured amongst students and faculty in academia. We also found kindness is most often remembered when people experience feeling safe and being acknowledged. Lastly, we provide evidence that reflecting on

experiences of kindness promotes positivity, with the potential to encourage conducting further acts of kindness. Given the results of our studies, the Kindness Received and the Kindness Given scales can be used to measure self-perceived kindness received and conduction of acts of kindness amongst academics, students and faculty in higher education. The Kindness scales can also be a diagnostic tool for both practitioners and researchers alike, to identify what specific experiences are occurring or lacking. The Kindness measures can be incorporated into evaluations of academic community climate surveys surrounding anti-racism and inclusion or to assess experiences of restorative justice circles now being used to increase inclusive communication practices. Training and interventions on how to be kind and the power of kindness can be created to support a positive, socially connected professional environment. Together these findings provide direction for those developing tools, workshops, and interventions to improve social connection and inclusion amongst academics. This study contributes to broader research highlighting positive contextual factors that promote persistence of students, staff, and faculty in academia.

## Supporting information

**S1 Table. Percentage, frequency, and examples of dignity affirmations for all 10 categories identified in the written responses.**
(PDF)

**S1 File. Demographics, correlation, and qualitative raw data.**
(CSV)

**S2 File. Structural equation models raw data.**
(CSV)

**S3 File. Kindness scales.**
(PDF)

## Acknowledgments

We thank and acknowledge Janice Vong, Natalia Maldonado, Dr. Gregory Hancock, Dr. Paul R. Hernandez, and Dr. Shujin Zhong for their support with this study.

## Author Contributions

**Conceptualization:** K. Kanoho Hosoda, Mica Estrada.

**Data curation:** K. Kanoho Hosoda.

**Formal analysis:** K. Kanoho Hosoda.

**Methodology:** K. Kanoho Hosoda, Mica Estrada.

**Project administration:** K. Kanoho Hosoda.

**Supervision:** Mica Estrada.

**Writing – original draft:** K. Kanoho Hosoda.

**Writing – review & editing:** K. Kanoho Hosoda, Mica Estrada.

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
