## [Decision Letter · Decision Letter 0]

30 Jul 2024

PONE-D-24-15380The Influence of Kindness on Academics’ Identity, Well-being and StressPLOS ONE

Dear Dr. Hosoda,

Thank you for submitting your manuscript to PLOS ONE. After careful consideration, we feel that it has merit but does not fully meet PLOS ONE’s publication criteria as it currently stands. Therefore, we invite you to submit a revised version of the manuscript that addresses the points raised during the review process.

I enjoyed the manuscript and I only have a few comments for you to revise.  Both reviewers also made only a few suggestions. on line 314 one of you stats became a square and on line 472 please change the &  to the word and We look forward to your edits.

We look forward to receiving your revised manuscript.

Kind regards,

Mary Diane Clark, PhD

Academic Editor

PLOS ONE

Journal Requirements:

"The research reported in this paper was supported by NIH grants 3R01GM138700-01S1 and #K99GM151640. We thank and acknowledge Janice Vong, Natalia Maldonado, Dr. Gregory Hancock, Dr. Paul Hernandez, and Dr. Shujin Zhong for their support with this study."

"KH #K99GM151640

National Institutes of Health https://www.nigms.nih.gov/

ME #3R01GM138700-01S1 

National Institutes of Health https://www.nigms.nih.gov/

The funder did not play any role in the study design, data collection and analysis, decision to publish, or preparation of the manuscript."

**Additional Editor Comments:**

Than you for submitting this interesting article. The reviews find it an important manuscript but note some editorial issues that we need to take care of as PLOS ONE does not have a copy editor

Reviewers' comments:

Reviewer's Responses to Questions

**Comments to the Author**

1. Is the manuscript technically sound, and do the data support the conclusions?

Reviewer #1: Yes

Reviewer #2: Yes

2. Has the statistical analysis been performed appropriately and rigorously? 

Reviewer #1: Yes

Reviewer #2: No

3. Have the authors made all data underlying the findings in their manuscript fully available?

Reviewer #1: Yes

Reviewer #2: Yes

4. Is the manuscript presented in an intelligible fashion and written in standard English?

Reviewer #1: Yes

Reviewer #2: No

5. Review Comments to the Author

Reviewer #1: In this study, they examine how academics experience kindness in their institutions and to what extent experiences of kindness are related to academics' experiences of well-being, stress, and self-identification with the institution. Overall, it is a carefully crafted study both in content and style. It addresses a novel topic, which constitutes a contribution to the literature. However, some minor suggestions are recommended to enhance the article.

Background:

There is a lack of connection between line 51 and 52. While the thread of the paragraph becomes clear towards the end, it is not apparent at the beginning. To facilitate comprehension, I would suggest dividing each overarching objective into specific ones.

Method:

Regarding sample selection, it is indicated that 77% were working in academia while 23% were students. However, the theoretical framework does not address the power hierarchy that may exist in vertical and horizontal relationships and the differences that may exist in the variables studied.

Results:

While some descriptive data are provided in the method, I believe it is necessary to add a figure or table with the means and standard deviations of the variables under study.

Discussion:

If potential differences between workers and students are not going to be addressed, it could be included as a future line of research. This may make more sense than the topic of aggressions.

Reviewer #2: The article, with its profound insights into the impact of the pandemic on well-being in higher education, is not just a crucial read for educators, but a testament to their value in this discussion. Understanding the challenges their students and colleagues face and learning how to support them through acts of kindness is a vital aspect of their integral role.

The author needs to address the following items:

- Ensure to include a hyphen in the word well-being consistently

- Font type and font size need to be consistent (lines 166, 169, 209-210, 219, 250-251, 264, 350, 352-353, 355-356, 360, 370, 376, 390, 401, 429, 435-436, 464, 472, 477)

- Research question - why is it italics? (lines 94-95 & 130-132)

- Remove extra spaces (lines 137 & 353)

- The study stated that 21.4% were in STEM fields but did not include 78.6%. Who was part of 78.6%? The total needs to add up to 100%. (line 176)

- Ethnic Demographic Distribution - the total does not add up to 100% (lines 177-180)

- Table 2 (line 309): Highlight the rows and columns interchangeability so that they are readable, and remove the border at the last row, "Note:..."

- Line 386 - need to confirm whether it should be N=162 or N=182 (Overall, (N=162),...)

- The total does not add up to 100% (line 403)

- Figure 1 diagram - Paths labeled Dgx & Drx in the description should be different, not the same.

- Figure 2 diagram - the word Kindness Given should be labeled instead of Kindness Action

- Supplemental Table 1 - font type needs to be consistent throughout

6. PLOS authors have the option to publish the peer review history of their article (what does this mean?). If published, this will include your full peer review and any attached files.

Reviewer #1: **Yes: **Julia Sánchez-García

Reviewer #2: No

---

## [Author Response · Author response to Decision Letter 0]

10 Sep 2024

Dear PLOS ONE Editor and Reviewers, 

We would like to thank you for the thorough and constructive feedback provided on our manuscript titled “The influence of kindness on academics’ identity, well-being, and stress” (Manuscript ID: PONE-D-24-15380). We have carefully considered each of the comments and have revised the manuscript to address the concerns raised. We believe that these revisions have significantly improved the quality of our work, and we are grateful for the opportunity to make these improvements.

Please find below our detailed responses to the comments, organized by reviewer. 

Editor: 

1. Comment: Please ensure manuscript meets PLOS ONE style requirements. 

a. Response: We have carefully reviewed and applied PLOS ONE’s style templates.

b. Action Taken: All headers, tables, figures, and supplemental materials have been revised to meet the requirements.

2. Comment: Funding should not appear in the acknowledgements section

a. Response: The funding information was in both the acknowledgements section and the funding statement.

b. Action Taken: Funding information was removed from the acknowledgements section.

3. Comment: Please include your full ethics statement in the methods section of the manuscript, including the full name of the IRB committee. 

a. Response: The requested information is now included in the Methods section

b. Action Taken: The following was included in the Methods section: “The study was reviewed and approved by the University of California San Francisco Mount Zion Committee Institutional Review Board (#21-35884). Written informed consent was obtained from all study participants.” 

4. Comment: Please include captions for your supporting files at the end of your manuscript. 

a. Response: Added a caption at the end of the manuscript file. 

b. Action Taken: The following was added at the end of the manuscript in the Supporting Information section: “S1 Table. Percentage, frequency, and examples of dignity affirmations for all 10 categories identified in the written responses.”

5. Comment: Please review your reference list to ensure that it is complete and correct. 

a. Response: All references were checked for completeness.

b. Action Taken: Reference 23 was updated with final publication information: 

Estrada M, Hosoda KK. Leading with Heroic Kindness. Encyclopedia of Heroism Studies. Springer Cham; 2023. Available: https://doi.org/10.1007/978-3-031-17125-3

6. Additional Change

a. The author, K. Kanoho Hosoda, has started a new position at the University of Hawaiʻi at Mānoa. The author affiliation and contact information has been changed accordingly. 

Reviewer 1: 

1. Comment: There is a lack of connection between line 51 and 52

a. Response: The author agrees with the comment. 

b. Action Taken: The second clause was removed from the sentence in line 51. The two sentences now read: “A variety of definitions of an act of kindness exists in the literature [18,19]. The emphasis in these definitions is on the nature of the action done.”

2. Comment: The theoretical framework does not address the power hierarchy that may exist between those working in academia versus those who are studying in academia. 

a. Response: The author agrees there is a power hierarchy that is not addressed. 

b. Action Taken: The power hierarchy is address in the discussion section, see comment below. 

3. Comment: It is necessary to add a figure or table with means and standard deviations of the variables under study

a. Response: The author agrees that the addition of a table with means and standard deviations would improve the manuscript.

b. Action Take: Table 1. Summary of sample descriptive statistics was included. 

4. Comment: If potential differences between workers and students are not going to be addressed, it could be included as a future line of research. This may make more sense than the topic of aggressions.

a. Response: The author agrees that the differences between workers and students should be addressed in future work. 

b. Action Taken: The following was included in the discussion section of the manuscript: 

“We acknowledge that increased variability may exist in this sample of academics, which includes both higher education students and people who work in higher education from various fields of study. We recognize there is a power hierarchy between students, staff, and faculty members in higher education that may influence outcomes. While this study did not directly address the power hierarchy within higher education, future studies would benefit from comparing these different communities within higher education environments.”

Reviewer 2: 

1. Comment(s): Formatting discrepancies including: 1) consistency of the hyphen in well-being, 2) font size, 3) italicizing research question, 4) extra spacing

a. Response: Author acknowledges the errors in formatting

b. Action Taken: Paper was reformatted using PLOS ONE style requirements and copy edited for hyphen consistency, font size, italics, and spacing. 

2. Comment: The study stated that 21.4% were in STEM fields but did not include 78.6%. Who was part of 78.6%? The total needs to add up to 100%.

a. Response: Percentages did not add up due to rounding errors

b. Action Taken: All percentages and numbers were rounded to the nearest hundredth.

3. Comment: Ethnic Demographic Distribution - the total does not add up to 100% (lines 177-180)

a. Response: Percentages did not add up due to rounding errors

b. Action Taken: All percentages and numbers were rounded to the nearest hundredth.

4. Comment: Table 2 (line 309): Highlight the rows and columns interchangeability so that they are readable, and remove the border at the last row, "Note:..."

a. Response: We recognize the error and acknowledge highlighting will improve readability

b. Action Taken: Rows were highlighted interchangeably for the table, which is now Table 3. 

5. Comment: Line 386 - need to confirm whether it should be N=162 or N=182 (Overall, (N=162),...)

a. Response: We agree it the sample was unclear. 

b. Action Taken: We included the percentage and further explanation as follows: “Notably, 89.01% of the participants (N=162 of the 182 study participants) listed words...”

6. Comment: The total does not add up to 100% (line 403)

a. Response: The author does not agree with the comment, as the total adds up to 100. 

b. Action Taken: None. 41.98+22.22+12.96+11.73+11.11= 100

7. Comment: Figure 1 diagram - Paths labeled Dgx & Drx in the description should be different, not the same.

a. Response: We agree with the comment

b. Action Taken: Path labels were revised in the figure. 

8. Comment: Figure 2 diagram - the word Kindness Given should be labeled instead of Kindness Action

a. Response: We agree with the comment. 

b. Action Taken: The label Kindness Action was revised to Kindness Given.

9. Comment: Supplemental Table 1 - font type needs to be consistent throughout

a. Response: We agree the supplemental table needed to be revised.

b. Action Taken: Formatting including font type and size were revised. Additionally, all percentages were rounded to the nearest hundredth instead of the nearest tenth to maintain consistency throughout the paper. 

We express our sincere gratitude to the reviewers and the editor for your thorough and insightful comments on our manuscript. We appreciate the opportunity to improve the manuscript based on your feedback.

We hope that the revised manuscript now meets the standards of PLOS ONE and addresses all the concerns raised. However, if there are any additional questions or further revisions needed, we are more than willing to address them promptly.

Thank you again for your time and consideration. We look forward to your favorable response.

---

## [Editor Report · Decision Letter 1]

4 Oct 2024

The influence of kindness on academics’ identity, well-being and stress

PONE-D-24-15380R1

Dear Dr. Hosoda,

We’re pleased to inform you that your manuscript has been judged scientifically suitable for publication and will be formally accepted for publication once it meets all outstanding technical requirements.

Kind regards,

Mary Diane Clark, PhD

Academic Editor

PLOS ONE

Additional Editor Comments (optional):

Thank you for your edits. the topic is interesting and we wish you well in your future publications.
---

## [Editor Report · Acceptance letter]

10 Oct 2024

PONE-D-24-15380R1 

PLOS ONE

Dear Dr. Hosoda, 

I'm pleased to inform you that your manuscript has been deemed suitable for publication in PLOS ONE. Congratulations! Your manuscript is now being handed over to our production team.

Kind regards, 

on behalf of

Dr. Mary Diane Clark 

Academic Editor

PLOS ONE